# Differences in right-to-left vs left-to-right interventricular conduction times in patients indicated to cardiac resynchronization therapy

David Pospisil[1,2¶], Tomas Novotny[1,2☯], Jiri Jarkovsky[3☯], Barbora Farkasova[1,2☯], Milan Kozak[1,2], Lubomir Krivan[1,2], Jitka Vlasinova[1,2], Petr Kala[1,2], Milan Sepsi[1,2☯]*

1 Department of Internal Medicine and Cardiology, University Hospital Brno, Brno, Czech Republic, 2 Faculty of Medicine, Masaryk University, Brno, Czech Republic, 3 Faculty of Medicine, Institute of Biostatistics and Analyses, Masaryk University, Brno, Czech Republic

☯ These authors contributed equally to this work.
¶ This author is the main author.
* sepsi.milan@fnbrno.cz

**Data Availability Statement:** The data underlying this study are available from the Harvard Dataverse

## Abstract

### Introduction

Differences in conduction times from right ventricle to left ventricle and from left ventricle to right ventricle respectively were observed during biventricular devices implantation when changing pacing vector direction. In this article the phenomenon of interventricular conduction time differences is described and assessed in relationship to various clinical and electrophysiological parameters.

### Methods

In 62 consecutive patients (9 females) interventricular conduction times between right and left ventricle in both directions were measured during cardiac resynchronization therapy device implantation procedure. Complex pacing protocol was performed.

### Results

Investigated individuals was divided into 3 subgroups according to type of interventricular conduction pattern and statistically tested with various clinical data. Substantial differences in right-to-left vs left-to-right conduction times (> 5 ms, range 7–72 ms) were observed in 24 (39%) of all patients. They were more common in patients with dilated cardiomyopathy (20 of 38, 53%) compared to 4 (17%) of 24 patients with coronary artery disease (p = 0.011). The phenomenon occurred more often in hypertensive patients (p = 0.012). Other tested factors were nonsignificant.

### Conclusions

There are almost no data on this topic. The occurrence of conduction difference phenomenon is quite common in dilated cardiomyopathy while it is rare in coronary artery disease.

using the following URL: https://doi.org/10.7910/DVN/ARGMZ9.

**Funding:** This work was supported by the Ministry of Health, Czech Republic - conceptual development of research organization (FNBr, 65269705) and MUNI/A/1446/2019. The funders had no role in study design, data collection, and analysis, decision to publish, or preparation of the manuscript.

**Competing interests:** The authors have declared that no competing interests exist.

We assume the diffuse nature of the disease and the way of remodeling of myocardium play the main role. Knowledge of this phenomenon could be useful in personalized cardiac resynchronization therapy optimization.

## Introduction

There are series of unanswered questions in the field of cardiac resynchronization therapy and various known approaches how to describe interventricular conduction delays. Interventricular conduction delay (IVCD) is a descriptor of interventricular electrical dyssynchrony between right ventricle (RV) and left ventricle (LV). It is measured between pacing lead in RV and sensing lead in LV. There are studies which found direct proportional relationship between measure of IVCD and reverse remodeling [1] or responsiveness to CRT (2).

Differences in conduction times from right ventricle to left ventricle and from left ventricle to right ventricle respectively were observed by our group during biventricular devices implantation. In this paper we show detailed description of this phenomenon and its relationship to various clinical factors.

## Materials and methods

The investigated group consisted of individuals who fulfilled indication criteria to CRT system implantation according to ESC Guidelines. All of them signed informed consents and agreed with CRT implantation and periprocedural pacing protocol.

### Implantation procedure

As usually, right atrial (RA) bipolar lead was implanted into the appendage. Right ventricular (RV) bipolar lead was placed in RV apex or septum according to physician's decision. Left ventricular (LV) quadripolar (rarely bipolar) lead was implanted preferably in lateral or posterolateral position. The final decision of its placement resulted from several parameters evaluation—acceptable R wave voltage, pacing threshold and good vein anchorage position with no phrenic nerve stimulation tested at maximum output (10 V).

### Conduction measurement protocol

All measurements were done using Biotronik ICS 3000 Operation / Implant module (Biotronik GmbH & Co.KG Berlin Germany). Conduction times were measured automatically using "Conduction times" tool integrated into the ICS 3000 Pacing System Module application.

Pacing vectors between RV and LV leads were tested and conduction times measured. We used bipolar mode only. In case of using quadripolar LV lead it meant to pace RV and to measure conduction times in LV lead—bipolarly coupled proximal & middle proximal, middle proximal & middle distal and middle distal & rings consecutively. Then LV lead was paced and conduction to RV lead measured. The following LV quadrupolar lead couples were paced consecutively: proximal & middle proximal, middle proximal & middle distal and middle distal & distal ring. Thus, using a quadrupolar electrode it was possible to create 6 unique vectors compared to only two when using a bipolar electrode.

**Measurement protocol was divided into four parts**:

1. To assess dependency of particular pacing rings location to interventricular conduction time all above mentioned vectors were tested.

2. To rule out voltage dependency of interventricular conduction times an incremental voltage test was performed. Pacing pulse width was set to 0.4 ms. Lowest pacing voltage was rounded to the closest upper whole number value over recently measured pacing threshold, then increased with 1 V steps up to 10 V. Pacing rate was set to 90 BPM to avoid intrinsic activation. Each pacing phase was long enough to reach stable interventricular conduction times.

3. To assess the influence of pacing rate on interventricular conduction time duration an incremental pacing test was performed. Pacing was started at heart rate equivalent to the closest upper ten of the intrinsic heart rate and then increased in 10 BPM steps up to 140 BPM. Pacing voltage in the test was set to double value of pacing threshold. Each pacing phase was long enough to reach stable interventricular conduction times.

4. To assess interventricular conduction during natural ventricular activation the atrial lead was paced incrementally. Initial pacing rate was equal to intrinsic heart rate rounded up to the closest ten. Pacing rate was then increased by 10 BPM steps up to 140 BPM or reaching the Wenckebach point. Pacing voltage in the test was set to double value of pacing threshold. Each pacing phase was long enough to reach stable interventricular conduction times. Conduction times to both ventricular leads were measured during atrial pacing. Interventricular conduction time differences at particular pacing rates were computed.

An average value was computed from at least five consecutive conduction times.

## Statistical analysis

Numerical data are presented as mean ± standard deviation. Fisher's exact test was used for categorical variables and Mann/Whitney test for continuous variables (alpha = 0.05). Statistical software used: IBM SPSS 25.0.0.1 (IBM Corporation, 2018). For analysis, subjects group was divided to three parts based on computed difference value. Group "RV→LV≈LV→RV"–bidirectionally comparable conduction with differences ≤ ±5 ms (n = 39). Group "RV→LV>LV→RV"–negative difference value—faster conduction from the left ventricle to the right ventricle (n = 12) and Group "RV→LV<LV→RV"—positive difference value—faster conduction from the right ventricle to the left ventricle (n = 11).

Together with meeting requirements of Helsinki's declaration the Ethics committee of Brno University Hospital and Masaryk University approved the project design and related patient informed consent.

## Results

In the period from February 2015 to March 2017 sixty-two patients (9 females, 15%), were recruited. Clinical characteristics are shown in Table 1.

The patients were divided into 3 subgroups based on interventricular conduction times difference values. Fig 1 shows values of interventricular conduction times difference and its distribution in the study population.

The conductions were considered as similar if differences were ≤ ±5 ms (n = 39)–the Group "RV→LV = LV→RV". Faster conduction from the left ventricle to the right ventricle–(the Group "RV→LV>LV→RV") was observed in 12 patients and absolute values of interventricular conduction differences ranged from 7 to 72 ms. Faster conduction from the right ventricle to the left ventricle–(the Group "RV→LV<LV→RV") was observed in 11 patients and absolute values of interventricular conduction differences ranged from 6 to 32 ms.

In Fig 2 similar data are shown separately for patients with dilated cardiomyopathy (DCM, n = 38) and coronary artery disease (CAD, n = 24) etiology of heart failure.

**Table 1. Characteristics of investigated group.**

| | RV→LV ≈ LV→RV (N = 39) | RV→LV > LV→RV (N = 12) | RV→LV < LV→RV (N = 11) | p-value |
|---|---|---|---|---|
| **Gender—N (%)** | | | | |
| Females | 5 (12.8%) | 2 (16.7%) | 2 (18.2%) | 0,883 |
| Males | 34 (87.2%) | 10 (83.3%) | 9 (81.8%) | |
| **Age—mean (± SD)** | 66 ±9 | 62 ±9 | 62 ±8 | 0,708 |
| **Ejection fraction—mean (± SD) (%) Median (5–95 percentile)** | 27.4 ± 4.9; 30.0 (20.0; 35.0) | 28.3 ± 6.3; 30.0 (17.0; 35.0) | 28.6 ± 6.7; 30.0 (15.0; 35.0) | 0,535 |
| **NYHA Class—N (%)** | | | | |
| I | 1 (2.6%) | 1 (8.3%) | 0 (0.0%) | 0,658 |
| II | 15 (38.5%) | 6 (50.0%) | 4 (36.4%) | |
| III | 23 (59.0%) | 5 (41.7%) | 7 (63.6%) | |
| IV | 0 (0.0%) | 0 (0.0%) | 0 (0.0%) | |
| **Etiology—N (%)** | | | | |
| DCM | 19 (48.7%) | 11 (91.7%) | 8 (72.7%) | **0,011** |
| ICM | 20 (51.3%) | 1 (8.3%) | 3 (27.3%) | |
| **Rhythm at implantation—N (%)** | | | | |
| Sinus rhythm | 38 (97.4%) | 8 (66.7%) | 10 (90.9%) | 0,060 |
| Atrial fibrillation | 1 (2.6%) | 3 (25.0%) | 1 (9.1%) | |
| Ventricular escape rhythm | 0 (0.0%) | 1 (8.3%) | 0 (0.0%) | |
| **QRS complex duration—mean (± SD) (%) Median (5–95 percentile)** | 146.2 ± 22.0; 150.0 (90.0; 172.0) | 153.8 ± 20.8; 150.0 (110.0; 195.0) | 152.1 ± 14.3; 155.0 (120.0; 175.0) | 0,796 |
| **Type of bundle branch block—N (%)** | 0 (0.0%) | 0 (0.0%) | 0 (0.0%) | |
| RBBB | 4 (10.3%) | 0 (0.0%) | 1 (9.1%) | 0,516 |
| LBBB | 34 (87.2%) | 12 (100.0%) | 10 (90.9%) | |
| Other | 1 (2.6%) | 0 (0.0%) | 0 (0.0%) | |
| **Position of lead in right ventricle—N (%)** | | | | |
| Apex | 17 (43.6%) | 6 (50.0%) | 2 (18.2%) | 0,210 |
| Septum | 22 (56.4%) | 6 (50.0%) | 9 (81.8%) | |
| **Position of lead in left ventricle—N (%)** | | | | |
| Anterior | 3 (7.7%) | 0 (0.0%) | 0 (0.0%) | 0,099 |
| Lateral | 30 (76.9%) | 6 (50.0%) | 8 (72.7%) | |
| Posterolateral | 6 (15.4%) | 6 (50.0%) | 3 (27.3%) | |
| **Hypertension—N (%)** | 29 (74.4%) | 6 (50.0%) | 3 (27.3%) | **0,012** |
| **Diabetes mellitus—N (%)** | 11 (28.2%) | 4 (33.3%) | 1 (9.1%) | 0,297 |
| **Hyperlipoproteinemia—N (%)** | 19 (48.7%) | 4 (33.3%) | 4 (36.4%) | 0,555 |
| **Vascular brain disease—N (%)** | 3 (7.7%) | 0 (0.0%) | 0 (0.0%) | 0,238 |
| **Ischemic disease of lower extremities—N (%)** | 1 (2.6%) | 0 (0.0%) | 0 (0.0%) | 0,626 |
| **Chronic renal insufficiency—N (%)** | 8 (20.5%) | 2 (16.7%) | 0 (0.0%) | 0,111 |
| **Chronic hepatopathy—N (%)** | 2 (5.1%) | 1 (8.3%) | 0 (0.0%) | 0,506 |
| **Chronic obstructive pulmonary disease—N (%)** | 2 (5.1%) | 0 (0.0%) | 0 (0.0%) | 0,388 |

P value—statistical significance of differences among investigated groups; Fisher's exact test for categorical variables, Mann-Whitney test for continuous variables (alpha = 0.05). Group "RV→LV≈LV→RV"–bidirectionally comparable conduction with differences ≤ ±5 ms; Group "RV→LV>LV→RV"–faster conduction from the left ventricle to the right ventricle; Group "RV→LV<LV→RV"–faster conduction from the right ventricle to the left ventricle; SD—Standard Deviation; NYHA—New York Heart Association; DCM—Dilated cardiomyopathy; CAD—Coronary Artery Disease; RBBB—Right bundle branch block; LBBB—Left bundle branch block;

Interventricular conduction differences were significantly higher in DCM subgroup compared to CAD subgroup (11 ±13 ms vs 3 ± 4 ms, p = 0.01). If difference up to 5 ms was considered normal, then marked conduction difference was observed in 20 (53%) of 38 patients with

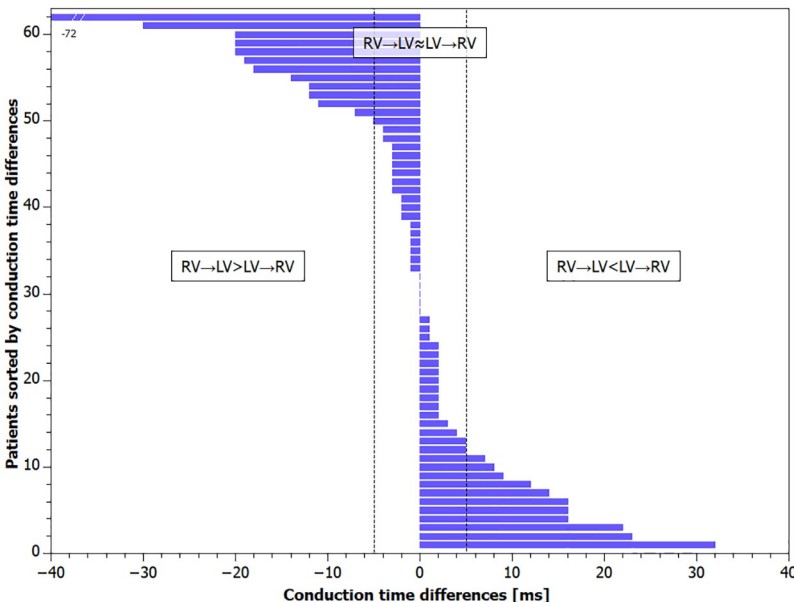

**Fig 1. Differences in interventricular conduction times sorted by differences value (whole study group N = 62).** Dashed line define group borders: Group RV→LV≈LV→RV—bidirectionally comparable conduction with differences ≤ ±5 ms (n = 39). Group RV→LV>LV→RV—negative difference value—faster conduction from the left ventricle to the right ventricle (n = 12) and Group RV→LV<LV→RV—positive difference value—faster conduction from the right ventricle to the left ventricle (n = 11). * Conduction time difference was -72 ms.

DCM vs 4 (17%) of 24 patients with CAD. Presence of interventricular conduction difference is statistically significantly related to cardiomyopathy etiology (p = 0.011).

Hypertension was more common in the Group A compared to B and C (74.4% vs 50% and 27.3%, p = 0.012).

Relationships to other clinical parameters (including QRS duration, ejection fraction or bundle branch block type) were not statistically significant.

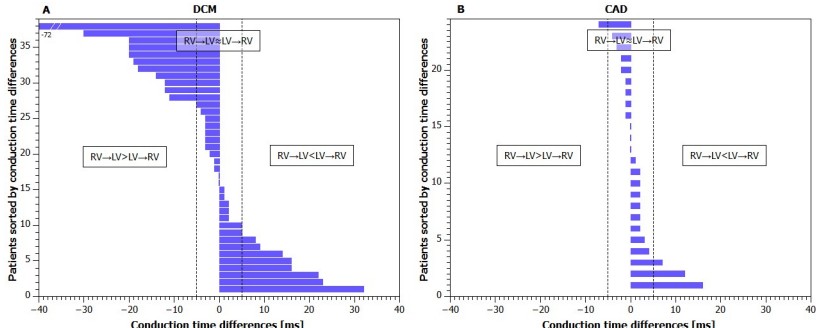

**Fig 2. Panel A: Interventricular conduction time differences in patients sorted by its value in DCM (N = 38) group.** Dashed line defines DCM subgroup borders. See Fig 1 for groups definitions. DCM patients belongs to: Group RV→LV≈LV→RV (N = 19), group RV→LV>LV→RV (N = 11), group RV→LV<LV→RV (N = 8). Panel B: Interventricular conduction time differences in patients sorted by its value in CAD (N = 24) group. Dashed line defines CAD subgroup borders. CAD patients belongs to: Group RV→LV≈LV→RV (N = 20), group RV→LV>LV→RV (N = 1), group RV→LV<LV→RV (N = 3).

## Discussion

In our study we assessed differences in conduction times from RV to LV and from LV to RV respectively in patients indicated for CRT. Pronounced differences (often up to 35 ms) were observed in substantial part of patients with DCM, while this phenomenon was much less common in patients with CAD.

It is well known that pathologic delay between activation of RV and LV during sinus rhythm with spontaneous conduction exists in heart failure patients with interventricular conduction disturbances. It is a descriptor of interventricular electrical dyssynchrony between right ventricle (RV) and left ventricle (LV). Several studies found direct proportional relationship between measure of IVCD and reverse remodeling [1] or responsiveness to CRT [2]. Interestingly, all these studies worked with right to left conduction times. According to our knowledge there are data neither on conduction times in the opposite direction, i.e. from LV to RV, nor on differences between both directions.

There was no notable difference in conduction time duration among any pacing vectors in concordant direction, at least at given distance between the proximal and distal rings of the LV lead. In other words, conduction times were similar throughout all the vectors in the same patient. Moreover, interventricular conduction times did not differ neither in relationship to pacing voltage nor to pacing rate (see Fig 3). There are no existing trials to compare these results with.

As mentioned above differences in conduction times in both directions were present in 53% of DCM patients compared to only 17% of CAD patients. Moreover, the absolute values of the differences were much higher in the DCM group. The reason is unclear. We can hypothesize that the particular way of remodeling of myocardium in the particular disease play the main role in the conduction differences. Histopathologic examination of hearts from end stage of cardiomyopathy shows reactive (interstitial and perivascular) fibrosis prevailing among patients with idiopathic DCM [3]. In contrast, a greater prevalence of replacement fibrosis is present in patients with CAD [4]. This special distinct pattern of diffuse fibrosis in DCM suggests a more generalized dysfunction which leads to slower conduction through myocardium [5].

Observation of less frequent hypertension in patients with interventricular conduction difference phenomenon is likely a collateral chance finding.

There are several approaches of interventricular conduction assessment. Several authors equal to several approaches. The main thing is that interventricular delay, in general, has been proven as it has its role in left ventricular remodeling [6] or prediction of CRT clinical response many times [7,8,9]. Other studies worked with RV to LV measurements. Since the resynchronization therapy is based mainly on preexcitation of the left ventricle to obtain synchronous ventricular contraction, it can be assumed there will be a relation with LV—RV conduction parameters or presence of some type of the difference with clinical outcome as well.

Knowledge of interventricular conduction differences might have important clinical impact. They are simple to measure with common implantation equipment during implantation procedure. In default settings a CRT device is programmed to pace LV and RV simultaneously [10,11]. Studies of LV offset for CRT programming have been disappointing [12]. In this regard implication of LV-RV vs RV-LV conduction differences could lead to more consistent results. To verify this concept further investigation is needed.

## Limitations

The group of patients is small, nevertheless this is a pilot study. The proportion of DCM and CAD was not equivalent similarly to other studies [13,14]. Due to small numbers it was not possible to perform any sex related comparisons.

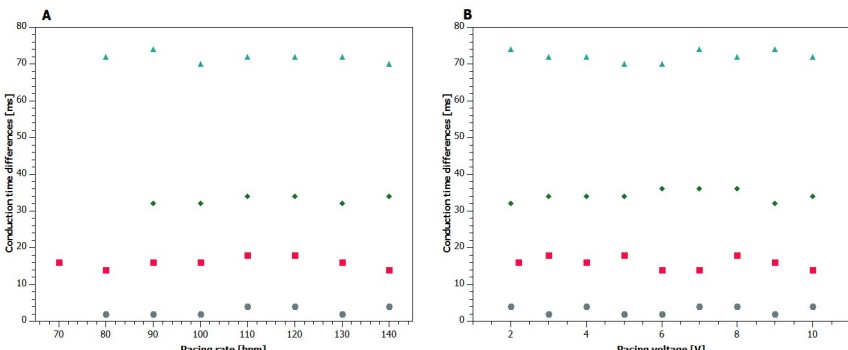

**Fig 3. Examples of pacing protocol results in four selected patients.** Panel A shows interventricular conduction time differences at different pacing rates. Panel B shows interventricular conduction time differences at different pacing voltages. Particular individual values do not substantially change neither in relationship to pacing rate nor to pacing voltage.

## Conclusion

The occurrence of differences in right-to-left vs left-to-right conduction times in patients indicated to cardiac resynchronization therapy is not rare. It was observed more often in patients with DCM compared to CAD. Knowledge of this phenomenon could be useful in optimization of ventricular timing in CRT patients.

## Author Contributions

**Conceptualization:** David Pospisil, Milan Sepsi.

**Data curation:** David Pospisil, Jiri Jarkovsky.

**Formal analysis:** Tomas Novotny, Jiri Jarkovsky, Milan Sepsi.

**Investigation:** David Pospisil, Milan Kozak, Lubomir Krivan, Jitka Vlasinova, Milan Sepsi.

**Methodology:** David Pospisil, Milan Sepsi.

**Project administration:** David Pospisil, Milan Sepsi.

**Supervision:** Petr Kala.

**Validation:** Jiri Jarkovsky.

**Visualization:** Barbora Farkasova.

**Writing – original draft:** David Pospisil.

**Writing – review & editing:** Tomas Novotny, Petr Kala, Milan Sepsi.

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
