## [Decision Letter · Decision Letter 0]

20 Dec 2019

PONE-D-19-32599

Differences in right-to-left vs left-to-right interventricular conduction times in patients indicated to cardiac resynchronization therapy

PLOS ONE

Dear  Dr. Sepsi

Thank you for submitting your manuscript to PLOS ONE. After careful consideration, we feel that it has merit but does not fully meet PLOS ONE’s publication criteria as it currently stands. Therefore, we invite you to submit a revised version of the manuscript that addresses the points raised during the review process.

We would appreciate receiving your revised manuscript by Feb 03 2020 11:59PM. To enhance the reproducibility of your results, we recommend that if applicable you deposit your laboratory protocols in protocols.io, where a protocol can be assigned its own identifier (DOI) such that it can be cited independently in the future. For instructions see: http://journals.plos.org/plosone/s/submission-guidelines#loc-laboratory-protocols

We look forward to receiving your revised manuscript.

Kind regards,

Giuseppe Coppola

Academic Editor

PLOS ONE

Journal Requirements:

Additional Editor Comments (if provided):

Kind author, your manuscript underwent three different reviewers; your paper is quite interesting but it does present some methodological problems.

We cannot accept as it is; you will find all the comments and criticisms underlined by reviewers.

We hope you could answer point by point to resubmit a revised version of your manuscript.

Reviewers' comments:

Reviewer's Responses to Questions

**Comments to the Author**

1. Is the manuscript technically sound, and do the data support the conclusions?

Reviewer #1: Yes

Reviewer #2: Partly

Reviewer #3: Yes

2. Has the statistical analysis been performed appropriately and rigorously? 

Reviewer #1: Yes

Reviewer #2: Yes

Reviewer #3: Yes

3. Have the authors made all data underlying the findings in their manuscript fully available?

Reviewer #1: Yes

Reviewer #2: Yes

Reviewer #3: Yes

4. Is the manuscript presented in an intelligible fashion and written in standard English?

Reviewer #1: Yes

Reviewer #2: No

Reviewer #3: Yes

5. Review Comments to the Author

Reviewer #1: Dear author,

the paper is well written and the topic is very interesting. The protocol is clear and precise while results appear hard to understand after a single reading and should be improved considering that they seem to be a long list of figures and tables description.

Moreover an important limitation of this study is the small number of enrolled patients.

At least, have you any data about response to resynchronization according to difference between inter-ventricular delay?

Regarding this topic I would like to suggest this paper regarding QRS index that can be considered a surrogate of reduced interventricular conduction delay: Magnitude of QRS duration reduction after biventricular pacing identifies responders to cardiac resynchronization therapy.

Coppola G, Ciaramitaro G, Stabile G, DOnofrio A, Palmisano P, Carità P, Mascioli G, Pecora D, De Simone A, Marini M, Rapacciuolo A, Savarese G, Maglia G, Pepi P, Padeletti L, Pierantozzi A, Arena G, Giovannini T, Caico SI, Nugara C, Ajello L, Malacrida M, Corrado E.

Int J Cardiol. 2016 Oct 15;221:450-5.

Reviewer #2: The paper from Sepsi and coll. is quite interesting but it does present some methodological problems.

1. The position of the RV lead could have had an impact on results. It is not simply a matter of how many RV leads have been implanted in a group or in another, but that changing the RV lead position IN THAT patients could have lead to completely different results and this impacts negatively on the study itself

2. The definition itself of "septal RV lead" is quite generical: which part of the septum? Hissian, parahissiam, septal RVOT? Again, this could have changed everything

3. The conclusions on correction on RV-LV lead delay (or viceversa; page 17, lines 213 - 214) and results of CRT is too simplicistic. FREEDOM Trial on use of QuickOpt did not demonstrate any benefit from this approach (it is almost the same concept, because the algorhitm measure the RV-LV delay during RV pacing and LV-RV delay during LV pacing and adjusts the VV interval so that the activation front is simultaneous)

Furthermore, I do not understand (Abstract, page 8, line 30) what does that < 5 ms mean: could the Authors make it clearer or at least explain in the revision letter the meaning of this statement.

Finally, the whole manuscript needs a deep english language revision because some passages are unclear.

Reviewer #3: Cardiac resynchronization therapy (CRT) is an established treatment for patients with heart failure (HF), impaired left ventricular (LV) function, and wide QRS complex.

Pospisil et al. described the phenomenon of interventricular conduction time differences and assessed relationship to various clinical and electrophysiological parameters. In fact the differences in conduction times from right ventricle to left ventricle and from left ventricle to right ventricle respectively were observed during biventricular devices implantation when changing pacing vector direction. In this study a pronounced differences (often up to 35 ms) were observed in substantial part of patients with dilatative cardiomyopathy (DCM), while this phenomenon was much less common in patients with coronary artery disease (CAD). The study is very interesting, presented in an appropriate fashion and are supported by interesting data. A limitation is represented by the small sample number (62 patients) to draw conclusions although the idea from a pathophysiological point of view is acceptable. Moreover it would be interesting to point out that other studies have also previously evaluated importance of CRT (see: “Non-responders to cardiac resynchronization therapy: Insights from multimodality imaging and electrocardiography. A brief review”, Doi: 10.1016/j.ijcard.2016.09.037). This would certainly increase the the article's quality.

In conclusion, given the overall work, I accept manuscript with minor revision concerning a small implementation of bibliography, reporting the aforementioned work and other.

6. PLOS authors have the option to publish the peer review history of their article (what does this mean?). If published, this will include your full peer review and any attached files.

Reviewer #1: No

Reviewer #2: Yes: Giosue Mascioli

Reviewer #3: No

---

## [Author Response · Author response to Decision Letter 0]

16 Jan 2020

PLoS ONE Brno, January 8, 2020

Editorial Office 

Dear editor,

Please find attached our revised manuscript called 

Differences in right-to-left vs. left-to-right interventricular conduction times in patients indicated for cardiac resynchronization therapy

by David Pospisil et al., (PONE-D-19-32599). 

Thank you for your invitation to submit a revised version of the manuscript. We highly appreciate the opportunity to respond to reviewer’s suggestions on the paper to clarify all questions and to meet PLoS ONE’s publication criteria. We look forward to hearing from you regarding our submission. We would be glad to respond to any further questions and comments that you may have.

Best regards

On behalf of all the authors

Milan Sepsi, MD, PhD

Department of Internal Medicine and Cardiology

University Hospital Brno, Masaryk University, Jihlavska 20, 625 00 Brno, Czech Republic

Phone: +420 777 227 346, Fax: +420 53223 2611, E-mail: sepsi.milan@fnbrno.cz

Response to Reviewer #1: 

Dear reviewer, 

thank you for your valuable time and useful contribution. We highly appreciate your inputs you given to improve the manuscript. We hope you will find our explanations and manuscript edits enough to your convenience. A point-by-point reactions to your comments and critics are written below directly into received texts using blue colored italic font. Attached file “Revised Manuscript with Track Changes” contains marked revisions made in the manuscript.

Dear author,

the paper is well written, and the topic is very interesting. The protocol is clear and precise while results appear hard to understand after a single reading and should be improved considering that they seem to be a long list of figures and tables description.

According to your proposition some corrections of the text have been applied. Nevertheless, the text of the “Results” section is quite short, substantial part of information is present in figures and their captions (which are inserted in the text of the manuscript immediately following the paragraph in which the figure is first cited).

Moreover, an important limitation of this study is the small number of enrolled patients.

We agree that the number is small and we have commented it in the limitations. Nevertheless, as a pilot study it can be sufficient to draw conclusions from a pathophysiological point of view. It is clear it deserves a further study on larger set of patients.

At least, have you any data about response to resynchronization according to difference between inter-ventricular delay?

Unfortunately, not yet. We are preparing a study to evaluate the phenomenon according to QRS width and responsiveness to CRT rigorously. 

Regarding this topic I would like to suggest this paper regarding QRS index that can be considered a surrogate of reduced interventricular conduction delay: Magnitude of QRS duration reduction after biventricular pacing identifies responders to cardiac resynchronization therapy.

Coppola G, Ciaramitaro G, Stabile G, DOnofrio A, Palmisano P, Carità P, Mascioli G, Pecora D, De Simone A, Marini M, Rapacciuolo A, Savarese G, Maglia G, Pepi P, Padeletti L, Pierantozzi A, Arena G, Giovannini T, Caico SI, Nugara C, Ajello L, Malacrida M, Corrado E.

Int J Cardiol. 2016 Oct 15;221:450-5.

We find suggested paper very interesting regarding our topic and it has been added to the manuscript. 

Response to Reviewer #2: 

Dear Dr. Mascioli, 

thank you for your valuable time and useful contribution. We highly appreciate your inputs you given to improve the manuscript. We hope you will find our explanations and manuscript edits enough to your convenience. A point-by-point reactions to your comments and critics are written below directly into received texts using blue colored italic font. Attached file “Revised Manuscript with Track Changes” contains marked revisions made in the manuscript.

Reviewer #2: The paper from Sepsi and coll. is quite interesting, but it does present some methodological problems.

1. The position of the RV lead could have had an impact on results. It is not simply a matter of how many RV leads have been implanted in a group or in another, but that changing the RV lead position IN THAT patients could have lead to completely different results and this impacts negatively on the study itself.

Of course, we considered that RV placement might have a significant impact to the interventricular conduction time and it obviously has. What we want to show is the fact that the differences of RV→LV and LV→RV conduction times are present regardless of RV lead placement in a given position.

2. The definition itself of "septal RV lead" is quite generical: which part of the septum? Hissian, parahissian, septal RVOT? Again, this could have changed everything. 

The information on septal vs apical position is based on implanting physician’s declaration. Unfortunately more detailed information on the lead position is not available. The aim of our manuscript is introduction of the phenomenon. Relationships to particular positions would require its study on larger group of patients.

3. The conclusions on correction on RV-LV lead delay (or viceversa; page 17, lines 213 - 214) and results of CRT is too simplicistic. FREEDOM Trial on use of QuickOpt did not demonstrate any benefit from this approach (it is almost the same concept, because the algorhitm measure the RV-LV delay during RV pacing and LV-RV delay during LV pacing and adjusts the VV interval so that the activation front is simultaneous)

The text was modified.

Furthermore, I do not understand (Abstract, page 8, line 30) what does that < 5 ms mean: could the Authors make it clearer or at least explain in the revision letter the meaning of this statement.

In original text, there was a mistake – thank you: we considered interventricular conduction times difference lower than 5 milliseconds (ms) as a bidirectionally comparable conduction (because of measurement error), so. statement < 5 ms corrected to “>5 ms”.

Finally, the whole manuscript needs a deep English language revision because some passages are unclear.

A complete grammar, language and terminology check was done by scientific manuscript editing service to fit the US-English standards. We include PDF file with certification by Proof.Reading.Service.Com Ltd.

Response to Reviewer #3: 

Dear reviewer,

thank you for your valuable time and useful contribution. We highly appreciate your inputs you given to improve the manuscript. We hope you will find our explanations and manuscript edits enough to your convenience. A point-by-point reactions to your comments and critics are written below directly into received texts using blue colored font. Attached file “Revised Manuscript with Track Changes” contains marked revisions made in the manuscript.

Reviewer #3: Cardiac resynchronization therapy (CRT) is an established treatment for patients with heart failure (HF), impaired left ventricular (LV) function, and wide QRS complex.

Pospisil et al. described the phenomenon of interventricular conduction time differences and assessed relationship to various clinical and electrophysiological parameters. In fact the differences in conduction times from right ventricle to left ventricle and from left ventricle to right ventricle respectively were observed during biventricular devices implantation when changing pacing vector direction. In this study a pronounced differences (often up to 35 ms) were observed in substantial part of patients with dilatative cardiomyopathy (DCM), while this phenomenon was much less common in patients with coronary artery disease (CAD). The study is very interesting, presented in an appropriate fashion and are supported by interesting data. A limitation is represented by the small sample number (62 patients) to draw conclusions although the idea from a pathophysiological point of view is acceptable. 

Moreover, it would be interesting to point out that other studies have also previously evaluated importance of CRT (see: “Non-responders to cardiac resynchronization therapy: Insights from multimodality imaging and electrocardiography. A brief review”, Doi: 10.1016/j.ijcard.2016.09.037). 

This would certainly increase the article's quality. In conclusion, given the overall work, I accept manuscript with minor revision concerning a small implementation of bibliography, reporting the aforementioned work and other.

We find the suggested paper very interesting regarding our topic and it has been added to the manuscript.

---

## [Editor Report · Decision Letter 1]

23 Jan 2020

Differences in right-to-left vs left-to-right interventricular conduction times in patients indicated to cardiac resynchronization therapy

PONE-D-19-32599R1

Dear Dr. Milan Sepsi,

We are pleased to inform you that your manuscript has been judged scientifically suitable for publication and will be formally accepted for publication once it complies with all outstanding technical requirements.

With kind regards,

Giuseppe Coppola

Academic Editor

PLOS ONE

Additional Editor Comments (optional):

Manuscript underwent language revision.

Authors have answerd to all reviewrs coments and criticisms when it was possible considering this manuscript a pilot study.
---

## [Editor Report · Acceptance letter]

28 Jan 2020

PONE-D-19-32599R1 

Differences in right-to-left vs left-to-right interventricular conduction times in patients indicated to cardiac resynchronization therapy 

Dear Dr. Sepsi:

I am pleased to inform you that your manuscript has been deemed suitable for publication in PLOS ONE. Congratulations! Your manuscript is now with our production department. 

With kind regards,

on behalf of

Dr. Giuseppe Coppola 

Academic Editor

PLOS ONE